# Acute Radiation Dermatitis Evaluation with Reflectance Confocal Microscopy: A Prospective Study

**DOI:** 10.3390/diagnostics11091670

**Published:** 2021-09-13

**Authors:** Juras Kišonas, Jonas Venius, Mindaugas Grybauskas, Daiva Dabkevičienė, Arvydas Burneckis, Ričardas Rotomskis

**Affiliations:** 1Department of Radiation Oncology, National Cancer Institute, LT-08660 Vilnius, Lithuania; mindaugas.grybauskas@nvi.lt (M.G.); arvydas.burneckis@nvi.lt (A.B.); 2Department of Neurobiology and Biophysics, Vilnius University, LT-01513 Vilnius, Lithuania; 3Medical Physics Department, National Cancer Institute, LT-08660 Vilnius, Lithuania; jonas.venius@nvi.lt; 4Biomedical Physics Laboratory, National Cancer Institute, LT-08660 Vilnius, Lithuania; ricardas.rotomskis@nvi.lt; 5Biobank, National Cancer Institute, LT-08660 Vilnius, Lithuania; daiva.dabkeviciene@nvi.lt

**Keywords:** acute radiation dermatitis, reflectance confocal microscopy, breast cancer

## Abstract

Background: During radiotherapy (RT), most breast cancer patients experience ionizing radiation (IR)-induced skin injury—acute radiation dermatitis (ARD). The severity of ARD is determined by a physician according to CTCAE or RTOG scales, which are subjective. Reflectance confocal microscopy (RCM) is a noninvasive skin imaging technique offering cellular resolution. Digital dermoscopy (DD) performed in conjugation with RCM can provide more information regarding skin toxicity. The purpose of this study is to create an RCM and DD features-based ARD assessment scale, to assess the association with CTCAE scale and possible predictive value. Methods: One hundred and three breast cancer patients during RT were recruited; every week, clinical symptoms of ARD (CTCAE scale) were evaluated and RCM, together with digital dermoscopy (DD), was performed. Results: According to RCM; after 2 RT weeks, exocytosis and/or spongiosis were present in 94% of patients; after 3 weeks, mild contrast cells (MMCs) were detected in 45%; disarrayed epidermis (DE) was present in 66% of patients after 4 weeks and in 93% after 5 weeks; abnormal dermal papillae (ADP) were present in 68% of patients after 5 weeks. The coefficients of RCM features (RCM_coef_) alone and together with dermoscopically determined erythema (RCM-ERY_coef_) were significantly associated with ARD severity grade. RCM_coef_ is a significant predictive factor for the clinical manifestation of ARD. Conclusions: RCM features of irradiated skin appear earlier than clinical symptoms, have a characteristic course, and allow the severity of ARD to be predicted.

## 1. Introduction

Whole breast radiotherapy (WBRT) after breast-conserving surgery (BCS) is the standard treatment for patients with breast cancer [1]. During RT, most breast cancer patients experience radiation-induced skin injury—acute radiation dermatitis (ARD)—and up to 30% of them suffer from a moderate or severe form of this side effect [2].

One of the most important risk factors for the severity of ARD is the IR dose to the skin. Definite erythema appears when a dose of more than 12 grays (Gy) of IR is delivered to the skin (2–3 weeks of RT). Dry desquamation can be diagnosed after 20 Gy (3–4 weeks of RT), and moist desquamation after a 30 Gy dose (>4 weeks of RT) [3].

In everyday practice, the severity of ARD is determined by a physician according to Common Terminology Criteria for Adverse Events (CTCAE) or Radiation Therapy Oncology Group (RTOG) scales. These scales are used to assess symptoms such as skin erythema, dry and moist desquamation, edema, bleeding, necrosis, and ulceration. The same scales are used in clinical studies investigating the efficiency of topical agents for ARD prevention and treatment. The progress of clinical studies searching for effective prevention and treatment for ARD has been slow [2], and one of the reasons is low-sensitivity and investigator-dependent diagnostics of skin changes.

Hypofractionated WBRT is increasingly being used, which reduces the severity of ARD [4], making it even more difficult to diagnose with classical methods. Consequently, the demand for more sensitive ARD diagnostic methods is increasing [5]. Moreover, ARD manifestation is observed in almost all RT patients, especially where higher IR doses are used, such as with head and neck cancer [6].

The histological features of ARD are not well understood, because it is rarely biopsied, mostly due to wound healing problems during RT. IR decreases collagen gene expression [7] and collagen quality, reduces proliferative ability, and causes molecular and subcellular changes [8], which can lead to wound healing complications for up to 30% of breast cancer patients who are biopsied during RT [9].

Noninvasive skin imaging techniques (optical biopsy) can replace histological examination, expand the knowledge of the pathogenesis, and improve the clinical diagnostics of ARD. Reflectance confocal microscopy (RCM), optical coherence tomography (OCT), and multiphoton microscopy (MPM) are the most frequently used in clinical trials and everyday practice optical biopsy techniques [10,11].

OCT can achieve up to 2–3 mm tissue depth but is limited to a relatively low image resolution of 1–15 µm [12]. MPM provides high lateral (<0.5 µm) and axial (1.0 µm) resolution [13]. However, it uses expensive short-pulsed lasers and is mostly used for research. On the other hand, RCM offers a similar resolution of 0.5–1.0 µm, uses a simple laser diode, and allows for the visualization of coetaneous structures with a resolution that is very close to that of light microscopy, thus performing a skin “optical biopsy” [14]. In addition to having a similar resolution, RCM also excels at convenient clinical application and is the most widely used skin imaging technique [15,16].

RCM employs near-infrared light targeting the “optical window” of the skin and uses a low power (below 20 mW) laser, which is below thermal threshold and causes no harm to the tissues, although the imaging depth is limited to the papillary dermis (200–300 µm). During imaging, light penetrates the tissue and illuminates an estimated 0.5 µm diameter point, and then the reflected light goes through a tiny pinhole, which does not allow the reflected light from the nearby tissue point to reach the detector. The contrast of RCM images depends on the reflectivity of the tissue. Structures with high refractive index (such as melanin and keratin) appear bright in RCM [17].

RCM is mostly used in the diagnosis of melanocytic lesions, such as nevi and melanoma [18,19], but is also useful for nonmelanocytic skin lesions [20,21,22], inflammatory diseases [23,24], and treatment monitoring [25,26]. RCM has been used for the evaluation of ARD skin changes, and it was determined that RCM can detect IR-induced skin changes, but the results were obtained from only six patients [27]. In one patient case study, it was also demonstrated that IR-induced skin changes in RCM images appear earlier than the occurrence of clinical symptoms [28].

A conventional RCM device is equipped with a dermoscope, which can provide more information regarding skin toxicity associated with RT [29]. The possibility of jointly employing digital dermoscopy (DD) with RCM for the better diagnosis of ARD is promising in terms of effective results of this combination in other fields [30,31].

Here, we present the results of a prospective, single center biomedical study including 103 breast cancer patients. The aim of this study is to create an RCM and DD features-based ARD assessment scale, to assess the association with CTCAE scale and possible predictive value. We found that the first signs of IR-induced skin lesions can be detected after the first week of RT; these violations have a typical course and can even predict the severity of the clinical symptoms of ARD.

## 2. Materials and Methods

### 2.1. Study and Patients

This prospective, open-label single center biomedical study (protocol No. II-2016-4) was approved (permission No. 158200-17-908-418) by the local bioethics committee in April 2017. Inclusion criteria were those aged >18 years, prescribed RT after BCS for local breast cancer, and exhibited good performance status (ECOG 0–2). Exclusion criteria were those who exhibited poor performance status (ECOG ≥ 3), undergone previous treatment with RT, or chemotherapy. A total of one hundred and ten patients with local breast cancer undergoing whole breast irradiation after BCS in the National Cancer Institute of Lithuania from April 2017 to February 2020 were recruited. A total of 103 patients with all RCM measurements were included in this paper from those who finished (108) the treatment.

### 2.2. Radiotherapy and Clinical ARD Assessment

Study patients were Caucasian women diagnosed with early-stage breast cancer and treated with BCS. According to the breast cancer treatment guidelines, whole breast RT was prescribed to all the patients. All patients were treated using the conventional 3D-RT technique. The prescribed dose to the breast planning target volume (PTV) was 50 Gy delivered in 2 Gy per fraction (fx), 5 fx per week.

A clinical assessment of ARD symptoms, according to the Common Terminology Criteria for Adverse Events (CTCAE) grading scale (Table 1), was performed before RT and once a week (every 5 fx). A total of three radiation oncologists separately evaluated radiation-induced skin reactions. The evaluations were combined, and the final degree of ARD was determined when at least two evaluators agreed.

### 2.3. Digital Dermoscopy

Dermoscopic images were taken before every RCM imaging with a VivaCam^®^ digital dermoscope (MAVIG GmbH, Munich, Germany) that uses a 2 MP CCD camera. These full-HD resolution images were used to assess the erythema (ERY) of the skin. A total of three radiation oncologists separately evaluated skin ERY according to a scale from 0 to 2, where 0 means “no erythema”, 1 means “faint erythema”, and 2 means “severe erythema”. The evaluations were combined, and the final grade was determined when at least two evaluators agreed.

### 2.4. Assessment of Skin Lesions with RCM

RCM images were taken with the commercially available VivaScope^®^ 1500 (MAVIG GmbH, Munich, Germany) medical device, which uses 830 nm wavelength, a 20 mW (max power) laser diode for tissue illumination, and a 30× magnification 0.9 NA water immersion objective for laser beam focusing onto the tissue and reflected light collection. The VivaScope^®^ 1500 provides a lateral resolution of 1.25 μm and an axial resolution of 5.0 μm. The size of individual images is 500 × 500 μm, with a frame rate of 9 images per second. The maximum mapped field is 8 × 8 mm^2^, the image resolution is 1024 × 1024 pixels, and the resultant magnification is 520×.

In the beginning of the imaging session, firstly, mosaic confocal reflectance images of 4 mm by 4 mm in the spinose-granular skin layer were taken. After that, 500 μm by 500 μm images every 5 μm up to 100 μm of depth were taken at three selected points. All of the images were evaluated by two observers. A lesion was graded as 1 if it was present in less than 50% of images and 2 if in more than 50%.

### 2.5. Statistical Analysis

Statistical analysis was used to describe characteristics of patients to compare different patient groups. For the descriptive statistics, scale variables were described by median and standard deviation and categorical variables by frequency of distribution. Ordinal logistic regression with sorted cases by subject and within-subject variables, and a Chi-square (X2) test, were used to determine the association between variables with similar radiotherapy fractions. Binary logistic regression was used to evaluate the association between variables with different radiotherapy fractions. The differences were considered statistically significant if a *p*-value was less than 0.05. Statistical analysis was performed using IBM-SPSS Statistics 21 (SPSS, Inc., Chicago, IL, USA).

## 3. Results

After the exclusion of a patient with missing RCM imaging, a total of 103 subjects were included in the further analysis, except for a few cases when moist desquamation appeared at the end of RT and RCM was discontinued.

All of the patients were females from 28 to 76 years old, with an average age of 57.1 years ± 8.97 (CI: 55.35–58.85). Study patient characteristics are shown in Table 2.

### 3.1. Clinical ARD Assessment

Clinical ARD assessment according to the CTCAE grading scale is presented in Table 3.

The patients in this study experienced the same clinical manifestation of ARD (Table 3) as other patients during the whole breast 3D-RT [32].

### 3.2. Dermoscopic Skin Imaging

Before RT, the control measurement of DD was performed on all patients. During treatment, dermoscopic images were taken after every 5 fx. Consecutive DD images revealed the progression of skin ERY during RT, which is depicted in Figure 1.

The ERY in the DD images was evaluated by three radiation oncologists according to a scale from 0 to 2, where 0 means “no ERY”, 1 means “faint ERY”, and 2 means “severe ERY”. In the left upper corner of Figure 1, the dermoscopic view of the patient’s skin before RT (0 fx) is demonstrated with no visual signs of ERY. All subsequent measurements were compared to the control. The example of the dermoscopic view of the faint ERY in a patient’s skin after 5 fx and 10 fx is demonstrated in the middle and right upper corner of Figure 1, and severe ERY after 15 fx, 20 fx, and 25 fx in the bottom of Figure 1. The results of skin ERY evaluations from all patients’ DD images are presented in Table 4.

Ordinal logistic regression showed that a higher degree of skin ERY, determined with DD, was associated (*p* < 0.0001) with a higher degree of ARD according to CTCAE scale.

In early fx (5 and 10), ERY using DD were diagnosed more frequently than according to the clinical evaluation (Table 3).

### 3.3. Skin Lesion Assessment with RCM

Before RT, the control measurement of RCM was performed on all patients. The representative view of the normal honeycomb pattern of epidermis before RT is shown in Figure 2a and the normal view of dermal papillae at the level of the dermo-epidermal junction (DEJ) in Figure 2b. All subsequent measurements were compared to the control.

The analysis of the first ten patients [33] revealed that spongiosis, exocytosis, and mild contrast cells (MCCs) are the early signs of radiation-induced skin injury, while disarrayed epidermis (DE) and abnormal dermal papillae (ADP) appear at the end of RT.

After 5 fx, the aggregates of round-to-polygonal cells corresponding to exocytosis (Figure 2c) were detected in 55.3% of all patients, but there were no alterations in the DEJ (Figure 2d). After 10 fx, an additional to exocytosis, the darker area relative to the surrounding epithelium of the stratum spinosum with intercellular spaces between larger than normal keratinocytes corresponding to spongiosis is demonstrated in Figure 2e, but still no alterations in the DEJ (Figure 2f). After 15 fx, the first signs of disarrayed epidermis (DE) were present in some patients’ RCM images (Figure 2g). Simultaneously, at the level of DEJ a single, aggregates or diffuse round-to-polygonal, mildly refractive cells (MCCs) (Figure 2h) corresponding to inflammatory cells were detected. Exocytosis and spongiosis are the earliest and nonspecific symptoms characteristic for other inflammatory conditions as well [17], while MMCs are considered as the “hallmark” of IR-induced skin injury [3]. These three RCM symptoms do not cause structural damage of the skin. The harm to skin structure can be diagnosed when the normal honeycomb-like architecture at the level of the stratum spinosum is lost (disarrayed epidermis (DE)) or when bright papillary rims at the DEJ are absent (abnormal dermal papillae (ADP)). After 20 fx, broad DE (Figure 2i) and MMCs in the level of the DEJ (Figure 2j) were present. After 25 fx, extensive DE (Figure 2k) and ADP at the level of the DEJ (Figure 2l) could be detected.

The manifestations of RCM features of all study patients are presented in Table 5.

By comparing these results with the degree of ARD (Table 3), we can clearly see that RCM features appear earlier.

### 3.4. Clinical ARD Association with RCM and DD

In order to compare the microscopic signs of radiation-induced skin injuries (spongiosis, exocytosis, MMCs, DE, and ADP) with clinical ARD evaluation (CTCAE scale), the coefficient of RCM features was calculated (RCM_coeff_). Firstly, the sum of the RCM (RCM_sum_) features was calculated according to Formula (1) as follows:RCM_sum_ = S,Emax + MMC + DE + ADP(1)

Here, spongiosis and exocytosis were counted as one symptom when only the maximum value of one of these symptoms was included in further calculations (S,Emax). The values of other symptoms (MMCs, DE, and ADP) were taken from Table 5.

RCM_sum_ was combined with RCM_coeff_, according to symptom acuity. Nonspecific inflammatory changes such as low-grade spongiosis or exocytosis were assigned an RCM_coeff_ value of 0. RCM symptoms that were reversible and corresponded to non-structural changes (such as higher grade spongiosis or exocytosis and MMCs) were assigned an RCM_coeff_ value of 1. High-grade symptoms causing damage to the skin structure were assigned an RCM_coeff_ value of 2. RCM_coeff_ values depending on RCM_sum_ are presented in Table 6.

RCM_coef_ values of all study patients are presented in Table 7.

Ordinal logistic regression revealed that a higher RCM_coeff_ is associated (*p* < 0.0001) with a higher degree of ARD, according to the CTCAE scale.

In order to compare the combination of RCM (spongiosis, exocytosis, MMCs, DE and ADP) and DD (ERY) with clinical ARD manifestation (CTCAE scale), the coefficient of RCM and DD features was calculated (RCM-ERY_coeff_). Firstly, the sum of the RCM and DD (RCM-ERY_sum_) features was calculated according to Formula (2) as follows:RCM-ERY_sum_ = RCM_sum_ + ERY(2)

Here, RCM_sum_ is calculated as mentioned above (Formula (1)), and ERY is the degree of skin erythema according to DD from Table 5.

RCM-ERY_coeff_ can have values from 0 to 2 depending on RCM-ERY_sum_ (Table 8).

RCM-ERY_coef_ values of all of the study patients are presented in Table 9.

Ordinal logistic regression showed that a higher RCM-ERY_coeff_ is associated (*p* < 0.0001) with higher degree of ARD, according to the CTCAE scale.

A X^2^ test was used in order to compare clinical ARD assessment (CTCAE scale) with ERY, RCM_coeff_, and RCM-DD_coeff_ when measurements were performed after the same number of fx (Table 10).

After 20 RT fractions, the levels of ARD clinical grade (CTCAE scale) were significantly associated with the levels of both RCM_coeff_ (*p* = 0.038) and RCM-ERY_coeff_ (*p* = 0.008), as the frequency of the higher subgroup of CTCAE increased with the higher subgroup of RCM_coeff_ or RCM-ERY_coeff_, while after 25 fx, the ARD clinical grade (CTCAE scale) was significantly associated with the values of both RCM_coeff_ (*p* = 0.01) and ERY (*p* = 0.038).

The next question was whether RCM and DD features, determined in the beginning of RT (from 10 fx to 15 fx), could significantly distinguish CTCAE grade 1 from grade 2 determined at the end of RT (20 fx and 25 fx). For the same patient, an RCM_coeff_ value of 1 after 10 fx and 15 fx had a significantly higher odds ratio of grade 2 in the CTCAE scale after 20 fx compared to grade 1 (Table 11).

Similarly, for the same patient, an RCM_coeff_ value of 1, determined after 15 fx, had a significantly higher odds ratio of grade 2 according to the CTCAE scale after 25 fx compared to grade 1 (Table 12).

In addition, the AUC estimates for RCM_coeff_ after 10 fx and 15 fx were also significant regarding CTCAE after 20 fx and 25 fx (Table 9 and Table 10). Therefore, the results of this analysis showed that RCM_coeff_, determined at the beginning of RT, significantly distinguished between grades 1 and 2 of the CTCAE scale at the end of RT and could be considered as a promising predictive factor.

## 4. Discussion

ARD is a relevant clinical problem, but until now the efficacy of topical treatment has not been proven in large, randomized studies [2]. There is a lack of new methods that could improve the diagnostics of ARD and accelerate the development of new topical agents for the prevention and treatment of this side effect.

RCM is widely used for melanoma non-melanoma skin cancer diagnostics [34,35,36,37] and treatment management [38,39]. RCM as an optical biopsy method is also used for inflammatory skin diseases diagnostics and treatment monitoring [23,24,40].

The aim of this study was to create an RCM and DD features-based ARD assessment scale, to assess the association with CTCAE scale and possible predictive value. For this purpose, we recruited 103 breast cancer patients during RT and every week we evaluated the clinical symptoms of ARD (CTCAE scale) and performed RCM. Before every RCM, the full-HD image of the same skin region was taken with DD. Given that RCM can be difficult to apply in everyday practice, we also analyzed DD images (Figure 1) and compared them with clinical ARD manifestation (CTCAE scale).

To our knowledge, this is the first prospective clinical study with more than 100 patients analyzing radiation-induced skin injuries with RCM and DD.

The first and, so far, the only study analyzing dynamic skin changes during RT using RCM was performed by Vano-Galvan and colleagues [27]. In our study, we investigated 103 patients and performed RCM once a week, while the Vano-Galvan study was conducted with 6 patients and RCM imaging every three weeks.

In our investigation, we performed more than 600 imaging sessions with RCM and DD. We found that by using RCM, the first symptoms of radiation-induced skin injury could be detected very early. After 5 fx, exocytosis or spongiosis was detected for 65% of all patients, and after 10 fx, 94% of patients experienced IR-induced skin injury (Table 5). During clinical evaluation, however, an ARD degree of 1 was diagnosed for 13% of patients after 5 fx, and after 10 fx only 40% of all patients were diagnosed with an ARD degree of 1 or 2 (Table 3).

According to RCM (Table 5 and Figure 2), radiation-induced skin changes have a typical course determined with RCM. After 2 weeks of RT (10 fx), exocytosis and/or spongiosis was detected for 94% of patients; after 3 weeks (15 fx), inflammatory cells (MMCs) were detected in 45% of patients, and this number increased until the end of RT; after 4 weeks (20 fx), DE was detected in 66% of patients and increased to 93% at the end of RT; after 5 weeks (25 fx), ADP was detected in 68% of patients.

Radiation-associated epidermal atypia was the main finding in RCM for basal cell carcinoma patients when the coetaneous response to RT was imaged in vivo [41]. The authors of this study suggested that epidermal atypia is likely a sign of ARD, which corresponds well with the findings of our study showing that at the end of RT, the majority of patients had pronounced DE.

For the more detailed comparison of RCM and DD symptoms with the clinical assessment of ARD, we created coefficients based on RCM and DD imaging results.

Ordinal logistic regression showed that the higher degree of ARD, according to the CTCAE scale, is associated with higher values of RCM_coeff_, ERY, and RCM-ERY_coeff_ (Table 10). Comparing measurements after the same number of fx, the strongest association between RCM or DD features and the CTCAE scale was found at the end of RT. After 4 RT weeks, RCM_coeff_ and RCM-ERY_coeff_ were significantly associated with the CTCAE scale grade, while after 5 weeks, similar results showed RCM_coeff_ and ERY (Table 10).

The final step was to analyze RCM and DD features as possible predictive factors for the severity of ARD determined by the CTCAE scale. According to binary logistic regression, RCM_coeff_ determined in the beginning of RT (from 10 fx to 15 fx) showed significant predictive value to distinguish the CTCAE grade 1 from grade 2 determined in the end (20 fx and 25 fx) of RT (Table 11 and Table 12).

These results are promising for integrating RCM into clinical studies or in everyday practice for the early detection of IR-induced skin changes and severity prediction. However, performing RCM requires time, effort, and a qualified specialist, which could be major limitations for applying it in clinical practice. On the other hand, DD is much easier to perform and interpret. Therefore, DD images could be analyzed by a physician and RCM by employing telediagnostics [42].

One of the possible limitations of this study is that all of the patients were Caucasian women. Some studies show that people with white skin have more severe clinical reactions to irritants compared to those with black skin [43]. A similar study with black skin patients should be conducted.

As mentioned above, there is a lack of clinical studies focusing on RCM application for skin changes during RT, but there are similar investigations with laboratory animals. For example, MPM used for early-stage radiation skin injury in a mouse model showed that after IR exposure, epidermal cells and intercellular spaces became large and irregular [44]. These symptoms correspond well with exocytosis and spongiosis in our study.

It was demonstrated that OCT angiography could non-invasively detect IR-induced changes in mouse skin, such as skin thickening, the dilation of large blood vessels, and irregularity in vessel boundaries [45].

Based on the results of this study, we propose RCM and DD as a sensitive tool for evaluation of IR-induced skin changes. The next step should be to analyze the benefits of RCM and DD in randomized clinical trials investigating the efficiency of topical agents for ARD prevention or treatment.

## 5. Conclusions

RCM reveals radiation-induced skin changes at the cellular level with no harm for the patient and could improve the assessment of skin condition during RT. RCM features of irradiated skin appear earlier than clinical symptoms, have a characteristic course, and allow for a prediction of the severity of ARD.

## Figures and Tables

**Figure 1 diagnostics-11-01670-f001:**
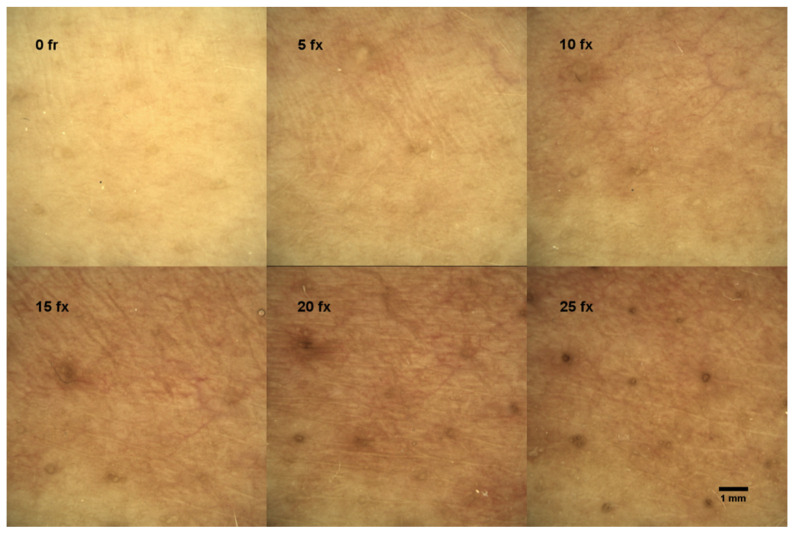
Dermascopic view of a single patient’s skin changes indicating the progression of skin ERY during radiotherapy; fx—radiotherapy fractions.

**Figure 2 diagnostics-11-01670-f002:**
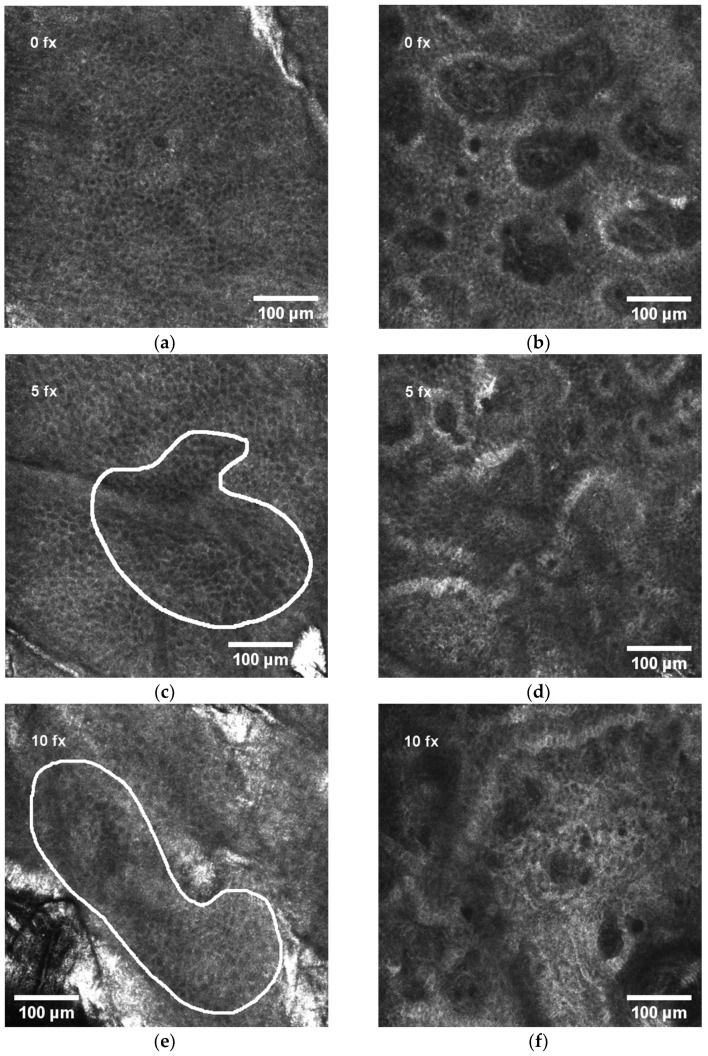
Dynamics of RCM features during radiotherapy: (**a**) normal honeycomb-like pattern of epidermis before radiotherapy; (**b**) normal view of dermal papillae in the dermo-epidermal junction (DEJ) before radiotherapy; (**c**) white line selection indicating exocytosis (aggregates of round-to-polygonal, mildly refractive cells) in epidermis after 5 radiotherapy fractions (fx); (**d**) normal view of dermal papillae in the DEJ after 5 fx; (**e**) white line selection indicating spongiosis (darker area relative to the surrounding epithelium of the stratum spinosum with intercellular spaces between keratinocytes larger than normal) and exocytosis in epidermis after 10 fx; (**f**) normal view of dermal papillae in the DEJ after 10 fx; (**g**) white line selection indicating the first signs of disarrayed epidermis (DE) after 15 fx; (**h**) white arrows indicating mild contrast cells (MMCs) in the DEJ after 15 fx; (**i**) white ovals indicating DE after 20 fx; (**j**) white arrows indicating MMCs in the DEJ after 20 fx; (**k**) white oval indicating large area of DE after 25 fx; (**l**) white circles indicating abnormal dermal papillae (ADP) after 25 fx.

**Table 1 diagnostics-11-01670-t001:** Common Terminology Criteria for Adverse Events (CTCAE) grading scale for acute radiation dermatitis.

Adverse Event	Grade				
	1	2	3	4	5
Radiation Dermatitis	Faint erythema or dry desquamation	Moderate to brisk erythema; patchy moist desquamation, mostly confined to skin folds and creases; moderate edema	Moist desquamation in areas other than skin folds and creases; bleeding induced by minor trauma or abrasion	Life-threatening consequences; skin necrosis or ulceration of full thickness dermis; spontaneous bleeding from involved site; skin graft indicated	Death

**Table 2 diagnostics-11-01670-t002:** Study patient characteristics.

Characteristic		Result
Patients (N)		103 (100%)
Gender	Female	103 (100%)
Age (y)	Mean, SD	57.1 ± 8.97
	Median	56.6
	Min–max	28–76
Stage	0	26 (25.2%)
	I	52 (50.5%)
	II	25 (24.3%)
HT	Yes	70 (68.0%)
	No	33 (32.0%)
Histology	DCIS	24 (23.3%)
	IDC	63 (61.2%)
	ILC	11 (10.7%)
	Other	5 (4.8%)
Radiotherapy	Dose 50 Gy in 25 fx	103 (100%)
	Technique (3D)	103 (100%)
	Dmean in 95% of CTV ± SD	48.8 ± 1.40
	HI (D2%–D98%)/D50% ± SD	0.44 ± 0.12

y—years, N—number, SD—standard deviation, min—minimum, max—maximum, HT—hormonotherapy, DCIS—ductal carcinoma in situ, IDC—invasive ductal carcinoma, ILC—invasive lobulal carcinoma, Gy—grays, Dmean—mean dose, CTV—clinical target volume, HI—homogeneity index, Dmax—maximum dose.

**Table 3 diagnostics-11-01670-t003:** Clinical evaluation of ARD according to CTCAE grading scale.

CTCAE Grade		0 fx	5 fx	10 fx	15 fx	20 fx	25 fx
ARD	0	103 (100%)	90 (87.4%)	61 (59.2%)	21 (20.4%)	7 (6.8%)	0
	I	0	13 (12.6%)	41 (39.8%)	80 (77.77%)	68 (66.0%)	43 (41.7%)
	II	0	0	1 (1.0%)	2 (1.9%)	28 (27.2%)	58 (56.3%)
	III	0	0	0	0	0	2 (1.9%)

CTCAE—Common Terminology Criteria for Adverse Events grading scale; fx—radiotherapy fractions, ARD—acute radiation dermatitis.

**Table 4 diagnostics-11-01670-t004:** The degree of skin erythema according to digital dermoscopy.

DD	deg	0 fx	5 fx	10 fx	15 fx	20 fx	25 fx
ERY	0	103 (100%)	71 (68.9%)	37 (35.9%)	16 (15.5%)	3 (2.9%)	2 (1.9%)
	1	0	31 (30.1%)	61 (59.2%)	78 (75.7%)	59 (57.3%)	28 (27.2%)
	2	0	0	3 (2.9%)	9 (8.7%)	39 (37.9%)	62 (60.2%)
	NP	0	1 (1.0%)	2 (1.9%)	0	2 (1.9%)	11 (10.7%)

DD—digital dermoscopy; ERY—erythema; deg—degree of erythema in DD imaged, where 0 means “no erythema”, 1 means “faint erythema”, and 2 means “severe erythema”; fx—radiotherapy fractions; NP—not performed.

**Table 5 diagnostics-11-01670-t005:** RCM features of all study patients.

RCMFeatures	Int.	0 fx	5 fx	10 fx	15 fx	20 fx	25 fx
Spongiosis	0	103 (100%)	75 (72.8%)	28 (27.2%)	9 (8.7%)	3 (3.0%)	1 (1.1%)
	1	0	28 (27.2%)	73 (70.9%)	80 (77.7%)	63 (62.4%)	50 (54.3%)
	2	0	0	2 (1.9%)	14 (13.6)	35 (34.7%)	41 (44.6%)
	NP	0	0	0	0	2 (1.9%)	11 (10.7%)
Exocytosis	0	103 (100%)	46 (44.7%)	8 (7.8%)	0	0	0
	1	0	57 (55.3%)	88 (85.4%)	59 (57.3%)	18 (17.8%)	7 (7.6%)
	2	0	0	7 (6.8)	44 (42.7%)	83 (82.2%)	85 (92.4%)
	NP	0	0	0	0	2 (1.9%)	11 (10.7%)
MMCs	0	103 (100%)	99 (96.1%)	86 (83.8%)	54 (52.4)	17 (16.8%)	2 (2.2%)
	1	0	4 (3.9%)	17 (16.5%)	47 (45.6)	77 (76.2%)	65 (70.7%)
	2	0	0	0	2 (1.9%)	7 (6.9%)	25 (27.2)
	NP	0	0	0	0	2 (1.9%)	11 (10.7%)
DE	0	103 (100%)	103 (100%)	102 (99.0%)	78 (75.7%)	34 (33.7%)	4 (4.3%)
	1	0	0	1 (1.0%)	24 (23.3%)	57 (56.4%)	43 (46.7)
	2	0	0	0	1 (1.0%)	10 (9.9%)	45 (48.9)
	NP	0	0	0	0	2 (1.9%)	11 (10.7%)
ADP	0	103 (100%)	103 (100%)	103 (100%)	102 (99.0%)	82 (81.2%)	29 (31.5%)
	1	0	0	0	1 (1.0%)	19 (18.8%)	58 (63.0%)
	2	0	0	0	0	0	5 (5.4%)
	NP	0	0	0	0	2 (1.9%)	11 (10.7%)

RCM—reflectance confocal microscopy; Int.—intensity of RCM features, where 1 means that the symptom is present in less than 50% of an image, and 2 in more than 50%; fx—fractions of radiotherapy; MMC—mild contrast cell; DE—disarrayed epidermis; ADP—abnormal dermal papillae; NP—not performed.

**Table 6 diagnostics-11-01670-t006:** RCM_coeff_ calculation from RCM_sum_.

RCM_sum_	RCM_coeff_
0–1	0
2–5	1
6–8	2

RCM_sum_—the sum of the reflectance confocal microscopy features, RCM_coeff_—the coefficient of reflectance confocal microscopy features.

**Table 7 diagnostics-11-01670-t007:** The values of RCM_coef_ of all study patients.

RCM	val	0 fx	5 fx	10 fx	15 fx	20 fx	25 fx
RCM_coef_	0	103 (100%)	97 (96.0%)	80 (77.7%)	41 (39.8%)	0	0
	1	0	4 (4.0%)	23 (22.3%)	60 (58.3%)	91 (90.1%)	46 (50.0%)
	2	0	0	0	2 (1.9%)	10 (9.9%)	46 (50.0%)
	NP	0	2 (1.9%)	0	0	2 (1.9%)	11 (10.7%)

RCM—reflectance confocal microscopy, RCM_coeff_—the coefficient of reflectance confocal microscopy features, val—value, fx—radiotherapy fractions, NP—not performed.

**Table 8 diagnostics-11-01670-t008:** RCM-ERY_coeff_ calculation from RCM_sum_ + ERY.

RCM-ERY_sum_	RCM-ERY_coeff_
0–1	0
2–5	1
6–10	2

RCM-ERY_sum_—the sum of the reflectance confocal microscopy and digital dermoscopy features, RCM-ERY_coeff_—the coefficient of reflectance confocal microscopy and digital dermoscopy features.

**Table 9 diagnostics-11-01670-t009:** The values of RCM-ERY_coef_ of all of the study patients.

RCM	val	0 fx	5 fx	10 fx	15 fx	20 fx	25 fx
RCM-ERY_coef_	0	103 (100%)	79 (78.2%)	32 (31.1%)	13 (12.6%)	0	0
	1	0	22 (21.8%)	70 (68.0%)	22 (85.4%)	66 (65.3%)	16 (17.4%)
	2	0	0	1 (1.0%)	2 (1.9%)	35 (34.77%)	76 (82.6%)
	NP	0	2 (1.9%)	0	0	2 (1.9%)	11 (10.7%)

RCM—reflectance confocal microscopy, RCM-ERY_coeff_—the coefficient of reflectance confocal microscopy and digital dermoscopy features, val—value, fx—radiotherapy fractions, NP—not performed.

**Table 10 diagnostics-11-01670-t010:** ARD degree according to CTCAE scales association with DD and RCM findings calculated by X^2^.

10 fx													
		ERY				RCM_coeff_				RCM-ERY_coeff_			
		0	1	2	*p*	0	1	2	*p*	0	1	2	*p*
CTCAE	0	26 (25.7%)	34 (33.7%)	0	0.062	46 (44.7%)	15 (14.6%)	0	0.721	24 (23.3%)	37 (35.9%)	0	0.059
	1	10 (9.9%)	27 (26.7%)	3 (3.0%)		33 (32.0%)	8 (7.8%)	0		7 (6.8%)	33 (32.0%)	1 (1.0%)	
	2	1 (1.0%)	0	0		1 (1.0%)	0	0		1 (1.0%)	0	0	
	3	0	0	0		0	0	0		0	0	0	
**15 fx**													
		**ERY**				**RCM_coeff_**				**RCM-ERY_coeff_**			
		**0**	**1**	**2**	** *p* **	**0**	**1**	**2**	** *p* **	**0**	**1**	**2**	** *p* **
CTCAE	0	3 (2.9%)	18 (17.5%)	0	0.151	8 (7.8%)	13 (12.6)	0	0.949	4 (3.9%)	17 (16.5%)	0	0.784
	1	13 (12.6%)	59 (57.3%)	8 (7.8%)		32 (31.1%)	46 (44.7%)	2 (2.5%)		9 (8.7%)	69 (67.0%)	2 (1.9%)	
	2	0	1 (1.0%)	1 (1.0%)		1 (1.0%)	1 (1.0%)	0		0	2 (1.9%)	0	
	3	0	0	0		0	0	0		0	0	0	
**20 fx**													
		**ERY**				**RCM_coeff_**				**RCM-ERY_coeff_**			
		**0**	**1**	**2**	** *p* **	**0**	**1**	**2**	** *p* **	**0**	**1**	**2**	** *p* **
CTCAE	0	1 (1.0%)	5 (5.0%)	1 (1.0%)	0.061	0	7 (6.9%)	0	0.038	0	7 (6.9%)	0	0.008
	1	1 (1.0%)	43 (42.6%)	23 (22.8%)		0	63 (62.4%)	4 (4.0%)		0	47 (46.5%)	20 (19.8%)	
	2	1 (1.0%)	11 (10.9%)	15 (14.9%)		0	21 (20.8%)	6 (5.9%)		0	12 (11.9%)	15 (14.9%)	
	3	0	0	0		0	0	0		0	0	0	
**25 fx**													
		**ERY**				**RCM_coeff_**				**RCM-ERY_coeff_**			
		**0**	**1**	**2**	** *p* **	**0**	**1**	**2**	** *p* **	**0**	**1**	**2**	** *p* **
CTCAE	0	0	0	0	0.038	0	0	0	0.01	0	0	0	0.078
	1	1 (1.0%)	19 (20.7%)	20 (21.7%)		0	27 (29.3%)	13 (14.1%)		0	11 (12.0%)	29 (31.5%)	
	2	1 (1.0%)	9 (9.8%)	41 (44.6%)		0	19 (20.7%)	32 (34.8%)		0	5 (5.4%)	46 (5.0%)	
	3	0	0	1 (1.0%)		0	0	1 (1.1%)		0	0	1	

ARD—acute radiation dermatitis, CTCAE—Common Terminology Criteria for Adverse Events grading scale, DD—digital dermoscopy, ERY—erythema, RCM—reflectance confocal microscopy, X^2^—Chi-square test, fx—radiotherapy fractions, *p*—*p*-value of significance, RCM_coeff_—reflectance confocal microscopy coefficient, RCM-ERY_coeff_—reflectance confocal microscopy and digital dermoscopy coefficient.

**Table 11 diagnostics-11-01670-t011:** Binary logistic regression of the diagnostic factors for CTCAE scale grades after 20 fx.

Univariate Analysis	Binary Logistic Regression		AUC	
Variable	OR (95% C.I.)	*p* Value	Area (95% CI)	*p* Value
RCMcoeff 10 fx: 1 0	4.83 (1.78–13.2) –	0.002 Ref.	0.64 (0.52–0.76)	0.022
RCMcoeff 15 fx: 1 0	14.5 (4.0–52.1) -	<0.001 Ref.	0.75 (0.65–0.84)	<0.0001

OR—odds ratio; AUC—area under the receiver operating characteristics curve; C.I.—confidence interval; Ref.—reference group; RCM-ERY_coeff_—reflectance confocal microscopy and digital dermoscopy coefficient; fx—radiotherapy fractions.

**Table 12 diagnostics-11-01670-t012:** Binary logistic regression of the diagnostic factors for CTCAE scale grades after 25 fx.

Univariate Analysis	Binary Logistic Regression		AUC	
Variable	OR (95% C.I.)	*p* Value	Area (95% CI)	*p* Value
RCMcoeff 10 fx: 1 0	5.0 (0.6–40.4) -	0.131 Ref.	0.59 (0.45–0.73)	0.24
RCMcoeff 15 fx: 1 0	8.8 (2.3–33.8) -	0.001 Ref.	0.74 (0.61–0.87)	0.002

OR—odds ratio; AUC—area under the receiver operating characteristics curve; C.I.—confidence interval; Ref.—reference group; RCM-ERY_coeff_—reflectance confocal microscopy and digital dermoscopy coefficient; fx—radiotherapy fractions.

## Data Availability

All data will be made available upon reasonable request to the corresponding author.

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
