# Peer review of "Acute Radiation Dermatitis Evaluation with Reflectance Confocal Microscopy: A Prospective Study"

_diagnostics, 2021, doi:10.3390/diagnostics11091670_

Round 1

Reviewer 1 Report

Kišonas et al. presented a systematic study of acute radiation dermatitis following ionizing radiation therapy of breast cancer patients. They observed the skin injuries resulted from ionization radiation using reflectance confocal microscopy (RCM), arguing that RCM is a novel and powerful tool for the prompt detection of such skin injuries. According to the authors, the relevance of the work relies on an unprecedented RCM-based study with more than 100 patient samples. 

As it is now, the manuscript can not be accepted:

i) the introduction does not define clearly the goal of the study.

ii) Although the authors claim that RCM is a powerful tool for skin investigations, they did not convincingly describe why neither the technique itself. How does RCM stand next to other microscopy techniques? What are the assets and weak spots of RCM for skin research?

iii) The authors did not provide a relevant state-of-the-art discussion.

iv) the results are obscure. Figure 1 appears out of the blue, and the authors gave no description nor explanation.

v) similarly, the description of Figure 2, which shows the best data of the study, is unclear, undetailed. I did not feel the effort of the authors to explain their results.

vi) I made some comments and remarks on the attached document. Consider reading/implementing them.

Author Response

Dear Reviewer,

Thank you for reweaving a manuscript entitled „Acute radiodermatitis evaluation with reflectance confocal microscopy: a prospective study“ by Juras Kišonas, Jonas Venius, Mindaugas Grybauskas, Daiva Dabkevičienė, Arvydas Burneckis and Ričardas Rotomskis which we are submitting to Diagnostics special issue Diagnostic Photoacoustic Imaging.

We appreciate your comments and remarks and try to implement them as much as it was passable in this short period of time.

Please, check the attached file for all changes.

Reviewer 2 Report

The author reports the use of reflectance confocal microscopy (RCM) as a noninvasive technique to study the IR induced skin injuries, acute radiation dermatitis (ARD), during radiotherapy treatment for breast cancer.

Suggest to revise the sentence, “The aim of this study … a new tool for early detection… during RT.” (line 66 – 67), because the RCM technique has been reported by the author and others in the literature for studying the injuries after radiotherapy (e.g., Radiat. Prot. Dosimetry. 182(1), 93-97, 2018). Moreover, the author should put more emphasis on the fact that this article distinguishes from others by employing the combination of RCM and DD for IR induced skin injuries in better prediction of ARD, based on a patient sample size of more than 100.

Please also give explanations for some of the abbreviations. As for example, DCIS, IDC, and ILC in table. Legends for these abbreviations should be provided.

The author may also consider to disclose some information about the skin color of the patients. Additionally, comments should be given if the skin color will affect the results of RCM, or DD. Please also indicate if any correction, or normalization has been performed for this large sample size due to the variation of skin color.

Author Response

(The authors gave the same response as above.)
